# Tissue-scale tensional homeostasis in skin regulates structure and physiological function

Shun Kimura [1,2], Ayako Tsuchiya [1], Miho Ogawa [1,3], Miki Ono [2], Nao Suda[1,4], Kaori Sekimoto [1,4], Makoto Takeo [1] & Takashi Tsuji[1✉]

Tensional homeostasis is crucial for organ and tissue development, including the establishment of morphological and functional properties. Skin plays essential roles in waterproofing, cushioning and protecting deeper tissues by forming internal tension-distribution patterns, which involves aligning various cells, appendages and extracellular matrices (ECMs). The balance of traction force is thought to contribute to the formation of strong and pliable physical structures that maintain their integrity and flexibility. Here, by using a human skin equivalent (HSE), the horizontal tension-force balance of the dermal layer was found to clearly improve HSE characteristics, such as the physical relationship between cells and the ECM. The tension also promoted skin homeostasis through the activation of mechanosensitive molecules such as ROCK and MRTF-A, and these results compared favourably to what was observed in tension-released models. Tension-induced HSE will contribute to analyze skin physiological functions regulated by tensional homeostasis as an alternative animal model.

[1] Laboratory for Organ Regeneration, RIKEN Center for Developmental Biology (CDB) and RIKEN Center for Biosystems Dynamics Research (BDR), Hyogo 650-0047, Japan. [2] ROHTO Pharmaceutical Co., Ltd., Osaka 544-8666, Japan. [3] Organ Technologies Inc., Tokyo 105-0001, Japan. [4] Meiji Seika Pharma Co., Ltd., Tokyo 104-8002, Japan. ✉email: takashi.tsuji@riken.jp

Environmental mechanical stresses, including gravity, shear stress, osmotic pressure and tension, are known to play important roles in the specific morphological and physical properties of tissues and organs[1–3]. Organ morphology and function, which are controlled by the extracellular matrix (ECM) and the traction-force balance in cells, are essential for maintaining organ homoeostasis[4]. The balance among extracellular forces exerted on cells by the ECM or neighbouring cells and the traction forces generated by cells themselves is termed tensional homoeostasis[5]. Cells sense the physical presence of tension from the surrounding environment via adhesion molecules and the actin cytoskeleton, and they respond to mechanical biological stimuli through mechano-transducers such as focal adhesion kinase (FAK), Yes-associated protein (YAP)/transcriptional coactivator with PDZ-binding motif (TAZ), and myocardin-related transcription factor A (MRTF-A)[6]. During embryogenesis, actomyosin-mediated tissue tension controlled by YAP is essential for organ shape formation and alignment of the organ systems, which is required for the proper three-dimensional (3D) body shape[1]. Actomyosin-mediated cellular tensional homoeostasis regulates tissue stiffness and β-catenin activation to induce epidermal hyperplasia and tumour growth[7]. Therefore, these spatiotemporal mechano-transduction processes and tensional homoeostasis play important roles in the regulation of cell functions, including myosin-based cell contraction, ECM production, cell migration, proliferation and differentiation[8] in organ and tissue morphogenesis and contribute to the performance of their specific functions[9].

The integumentary organ system (IOS), including skin and skin appendages such as hair, sebaceous glands, sweat glands, feathers, and nails, plays essential roles in waterproofing, cushioning, protecting deeper tissues, excreting waste and thermoregulation[10]. Alignment of components of the IOS, such as appendages, cells, dermal ECM such as collagen and elastic fibres, is regulated by internal tension-distribution patterns called Langer's cleavage lines, and they contribute to the formation of strong and pliable physical structures for the maintenance of their integrity and flexibility[11]. Tissue-scale tensions are known to involve mechano-sensors and mechano-transducers such as glycocalyx, lipid rafts/caveolin-1, cell adhesion (which is mediated by integrin, hemidesmosomes) and focal adhesion in the IOS morphogenesis process[12,13]. The alignment of various cells and structures is thought to be regulated by mechano-sensitive ion channels such as calcium-sensing receptors, transient receptor potential channels, connexions and Piezo channels[14], intercellular complexes of desmosomes or cadherin[15], and the actin cytoskeleton[16–18]. Defects in tensional homoeostasis and mechanical stress signalling involved in skin ageing, wound healing and disease[19] cause tension reduction through the degradation of the dermal ECM components, such as collagen and elastic fibre, which affect the progression of cellular dysfunction, including decreased ECM production and increased tissue protease production[20,21]. In scleroderma, MRTF-A, which is a central regulator of tissue fibrosis that connects mechanical cues and ECM remodelling, is activated in dermal fibroblasts[22]. Tensional homoeostasis is thought to play important roles in maintaining skin organ homoeostasis via mechanical stress signals.

3D human skin equivalents (HSEs) can partially reproduce physiological skin functions, such as proliferative capacity[23], ECM synthesis[24], cellular signalling and responses to various stimuli, and they are thought to be more useful models for drug discovery, disease modelling and basic skin research than two-dimensional monolayer cell cultures[25]. Recently, the application of HSE models as alternative animal experiments for safety assessment has been promoted in skin research and drug discovery[26,27]. To improve the HSE structures and functions, various studies on cell types, substrates, culture conditions, 3D bioprinting and Skin-on-Chip system have been reported[28–30]. Our recent studies provided a proof of concept regarding fully functional regeneration of tissues and organs, such as teeth, hair follicles, secretory glands and the IOS, that mimic the developmental processes of organogenesis[31–33]. However, it is expected to develop a more functional HSE compared to the present limited availability to study of skin functions as an alternative animal experiment to recapitulate the 3D anatomy of the IOS[28].

Here, we developed a HSE that maintains the ECM and cellular alignment in accordance with tension by reproducing tensional homoeostasis. We also found that tensional homoeostasis improves the physiological functions of HSE, including the production of the ECM by fibroblasts, the proliferation of epidermal keratinocytes and the responsiveness of cells to several functional molecules. We also found that reproduction of tensional homoeostasis induces changes in cell adhesion through integrin α2, and it achieves activation of mechanical stress signalling through nuclear translocation of MRTF-A. Our study suggests that skin tension balance regulates skin functionality via mechanical stress signals through epithelial–mesenchymal interactions.

## Results

**Development of a HSE exhibiting skin tensional homoeostasis.** Natural skin performs its typical functions under tensional homoeostasis. We first investigated the effect of tensional homoeostasis on the natural skin structure through the maintenance or release of skin tension (Fig. 1a). Tension-released skin was obtained from the resection of a part of contracted skin and included both epidermis and dermis, the reduction of interfollicular spaces, and vertical rearrangement of hair follicle compared with normal skin (Fig. 1b). Bell's model, which was constructed by using contracted collagen gel as dermal equivalents (DE) in a 7-day floating culture, is a well-known typical artificial skin model[23,34]. To investigate the roles of skin tension on skin structures and functions, we developed a tension homeostatic skin model (THS model) that reproduced traction-force balance in the lateral direction by clamping DE in a snap-well culture insert. DE are composed of two layers at different cell concentrations; because the upper layer has a high concentration, it will induce reciprocal interactions between epidermal cells and dermis fibroblasts. A tension-released skin model (TRS model) was generated by resection of the THS model from the wall of the culture insert (Fig. 1c). In all culture systems, HSE were placed in 6-well culture plates, and there was no contact between the epidermal surfaces and the medium (Fig. 1d, e). Histological analysis showed that Bell's model and the TRS model displays tissue contractile ability similar to that of tension-released natural skin (Fig. 1b, e). In contrast, in the THS model, the fibroblasts were oriented horizontally, and the epidermal cells were well aligned (Fig. 1e, middle). In the THS models, the epidermal cell layer was differentiated and had organized into epidermal tissue, the basal, spinous, granular and stratum corneum cell layers, within 5–7 days of culture in an air–liquid interface (Fig. 1f). After 7 days in the air–liquid interface culture, the localization of keratinocyte differentiation markers, including Keratin 5 (KRT5), Keratin 10 (KRT10), FLG and Claudin 1 (CLDN1), was observed at the epidermal layer, which was similar to that of natural skin (Fig. 1g). In the dermis layer, the localization of Type 1 Collagen (COL1) was detected in the dermal ECM (Fig. 1g). By contrast, Type 3 collagen (COL3) was not detected in the THS model which was constructed by using adult-derived fibroblasts, because this collagen is known highly produced and constructed during embryogenesis compared to adult stage[35]. Collagen deposition was clearly observed in the upper layer of the dermis than that in

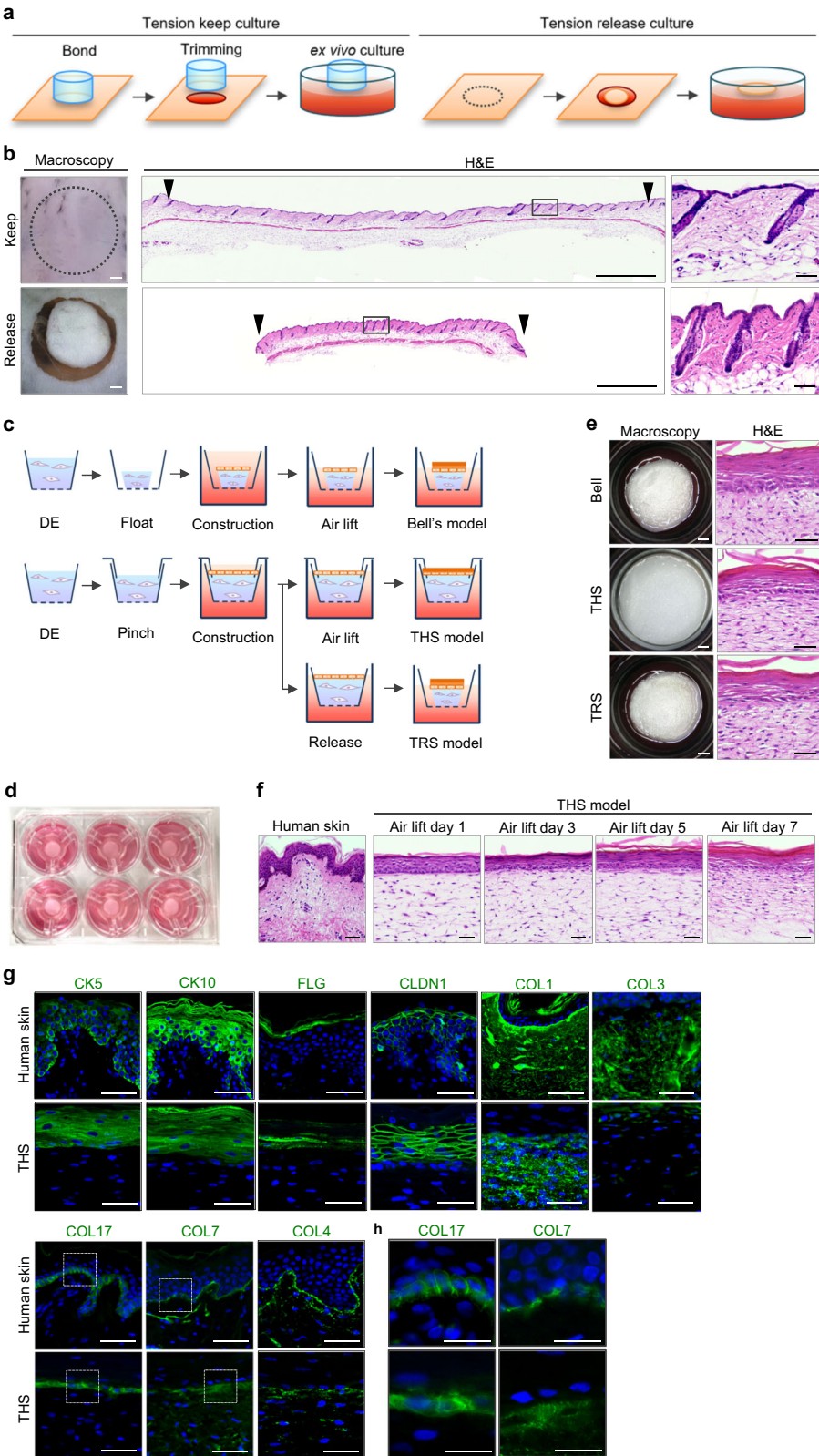

**Fig. 1 Construction of skin equivalent with skin tension homoeostasis. a** Schematic representation of ex vivo culture methods maintaining or releasing skin tensional homoeostasis. **b** Histological analyses of *ex vivo* cultured skin maintaining (upper columns) or releasing (lower columns) tensional homoeostasis. Macroscopic images (left panels) are shown. Scale bars, 1 mm. H&E staining (centre panels) and higher magnification images in each box area are shown (left panels). Scale bars, 1 mm or 50 μm. **c** Schematic representation of the construction methods of HSEs. **d** Images of the THS model. **e** Macroscopic (left) and H&E staining (second panels from the left) images of each model. Scale bars, 1 mm or 50 μm. **f** H&E staining of human skin and the THS model after 1, 3, 5 and 7 days of culture at the air–liquid interface. Scale bars, 50 μm. **g, h** Immunohistochemical analyses of human skin (upper columns) and the THS model (lower columns) after 7 days of culture at the air–liquid interface. Scale bars, 50 μm.

the lower layer. In the basement membrane zone, the slightly broad localizations of Type 17 collagen (COL17), Type 4 collagen (COL4) and Type 7 collagen (COL7), which are each structural component of hemidesmosomes, basal lamina and anchoring fibrils, respectively, were observed compared to human skin (Fig. 1g, h).

**The effect of reproduced tensional homoeostasis on dermal tissue alignment**. The alignment of natural skin ECM components, such as collagen and elastic fibres, is configured along an internal tension-distribution pattern, which is called Langer's cleavage lines[11]. We next investigated the effect of tensional homoeostasis on dermal tissue alignment by histological and mathematical analysis of nuclear morphology and the distribution of Type 1 collagen fibres. In natural skin with tensional homoeostasis, the shape of the dermal cell nuclei was horizontal relative to the direction of the traction force, and Type 1 collagen fibres were also distributed in the direction of the traction force (Fig. 2a upper and left columns). In contrast, in the tension-released natural skin, the shape of the nucleus and the internuclear distance became round and shortened, respectively, and the alignment of Type 1 collagen fibres was disrupted (Fig. 2a lower columns). The angle was measured as the orientation of the nuclear morphology by outlining the nuclear-counterstained images and fitting an ellipse into individual segmented nuclei in the programme FIJI; for the natural skin, the angle was also parallel relative to the direction of the traction force (Fig. 2a upper and middle column). In addition, the alignment of Type 1 collagen fibres, which was mathematically evaluated by FFT analysis, was found to be horizontal relative to the direction of the traction force (Fig. 2a upper and right column). In contrast, in the tension-released natural skin, it was mathematically shown that both the orientation of the nuclei and the alignment of the collagen fibres were disrupted as a response to the decrease in tensional homoeostasis (Fig. 2a lower column). Next, we performed the same analyses for each HSE model. The THS model was the only one where the orientation of the nuclei and the Type 1 collagen fibres was found to be horizontal in relation to the direction of tension (Fig. 2b middle columns). By contrast, both in Bell's and TRS models, the nuclear morphology, angle and collagen fibre alignment were similar to those of the tension-released natural skin (Fig. 2b upper and lower columns). Furthermore, to investigate the cell shape and the cytoskeleton in these THS models by using 3D analysis, the cell orientation was detected by plasma membrane staining in the THS model. The orientation was also aligned in the horizontal direction by the traction force, but this was not observed in Bell's and TRS models, in which the cell arrangement was disturbed (Fig. 2c). The actin fibre structure detected by phalloidin in both Bell's and TRS models was not horizontal compared to that in the THS model, which was aligned horizontally in relation to the direction of the tension (Fig. 2d). These results indicate that tensional homoeostasis plays important roles in cell shape and nuclear morphology through the rearrangement of cellular matrices (Fig. 2e).

**The THS model can evaluate skin barrier function**. Various HSE models, including Bell's model without tension and epidermal layer models, are used to evaluate skin barrier function and potential hazardous chemicals such as skin irritation[36]. We next examined whether the THS model could be applied to analysing skin barrier functions according to previously performed procedures for these assays[37]. The surface structure of the THS model epidermis was disrupted by the addition of SDS, which is a typical skin irritant, and the response was dose-dependent (Fig. 3a). A cytotoxicity assay using the THS model

also showed the cytotoxic effect of SDS at an IC50 value of 0.28% (Fig. 3b). In the assays of transepidermal water loss (TEWL) and transepidermal electrical resistance (TEER), which were performed according to a known procedure[38,39], the skin barrier in the THS model was clearly disrupted by administration of 0.20% SDS (Fig. 3c, d). To visualize and quantitatively evaluate molecular absorption, we performed a topical administration assay using fluorescein (Fig. 3e, f). Compared to the results of PBS exposure on the skin of the THS model, in which fluorescein was present in the upper epidermis layer (Fig. 3e, f, left), SDS treatment drastically increased the diffusion of fluorescein into the epidermis layer (Fig. 3e, f, right). These results indicate that our developed THS model can be used in known barrier function assays in a fashion that is similar to conventional HSE models.

**Reproduced tensional homoeostasis promotes skin self-organization and drug responsiveness of the HSE**. Tensional homoeostasis is well known to regulate not only tissue structure but also cell and tissue function. We next investigated whether there was an improvement in gene and protein expression related to skin functions, including genes functioning in the ECM and/or skin homoeostasis, following reproduction of tensional homoeostasis. We first examined the relative gene expression of genes related to the dermal ECM. These genes, *COL1A1*, *COL1A2*, *FBN1* and *ELN*, were drastically induced in the THS model compared to their expression in the Bell's and TRS models (Fig. 4a). The dermal ECM degradation-related gene *MMP1* was downregulated only in the THS model (Fig. 4a). Unusually, the formation of Type 1 collagen fibres was clearly observed in dermis layers, and its orientation was linked to the direction of the traction force in the THS model, which was not observed in the Bell's and TRS models (Fig. 4b). These results clearly indicate that traction-force balance in the THS model plays an essential role in the gene expression of homoeostasis-related genes in the skin.

We also analysed the effect of tensional homoeostasis on epidermal tissue structures by immunohistochemical analysis. In the HSE models, the epidermal structures and the dermis layer were improved[40]. We also observed the following expression characteristics and confirmed that there was no difference between THS, TRS and Bell's models: COL7 and COL17 was expressed in the basement membrane zone, KRT5 in the basal layer, KRT10 in the spinous to stratum corneum cell layers, FLG in the granular layer, and CLDN1 in the spinous and granular layers in all HSE models (Fig. 4b). In natural skin, drugs or nutrients are delivered to cells via percutaneous absorption or via the circulatory system. Skin cells are known to respond directly or through the interaction between epidermal and dermal layers. KGF is a key molecule that forms skin structure through epithelial–mesenchymal interactions[41]. The gene expression levels of *KGF* were observed to be higher in the THS and TRS models than they were in Bell's model (Fig. 4c). Furthermore, the number of Ki67-positive cells was significantly increased in THS epidermis compared with that of the Bell's and TRS models (Fig. 4d).

To investigate the tensional homoeostasis as a function of skin drug responsiveness by using these HSE models, we examined two drug administration models: the intravenous administration model (IAM) and the topical application model (TAM) (Fig. 4e). It is well known that all-*trans*-retinoic acid (ATRA) improves chronologically aged and photoaged skin by promoting epidermal turnover and synthesis of dermal collagen[42,43]. Under unstimulated conditions in the THS and TRS models, the mRNA levels of *COL1A1* was observed to be expressed at two-fold higher levels than it was in Bell's model, and its expression was drastically induced only in the THS model by using the IAM (Fig. 4f, left).

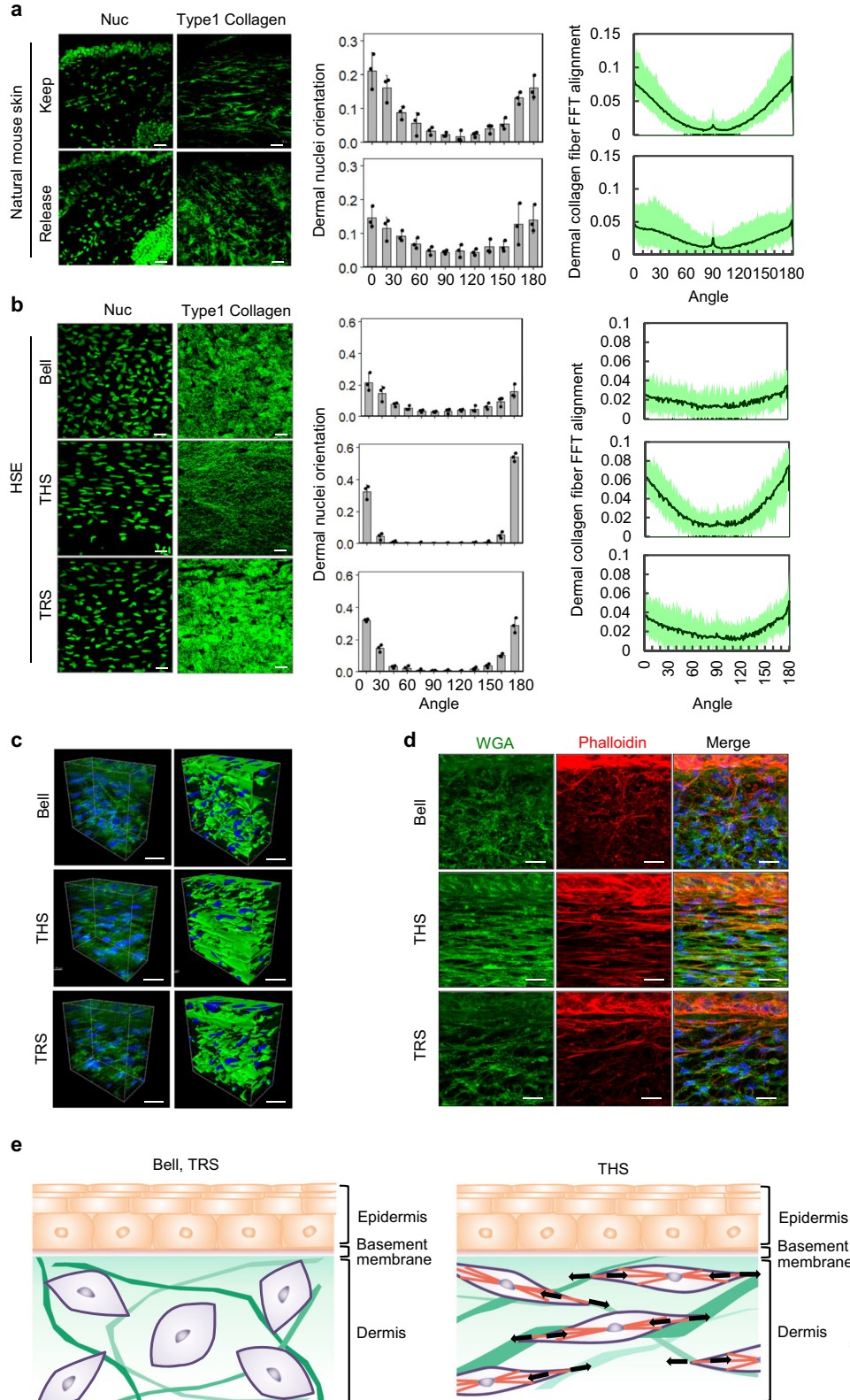

The gene expression of *HAS3*, which is produced in the epidermis, was significantly induced in the THS model when using the IAM, although *HAS3* expression was clearly induced in all HSE models (Fig. 4f, right). In the THS model, the induction of *HAS3* expression was observed when using both the IAM and TAM (Fig. 4g). These results indicate that several skin functions, including gene and protein expression, are involved in both the

steady-state and the response to drug treatment, as revealed by the reproduction of traction balance in the HSE model; these results suggest that epithelial and mesenchymal interactions are strengthened in the culture model.

We further evaluated drug responsiveness to TGF-β, which is known to play roles in tissue repair via the production and remodelling of ECMs such as COL1A1, COL1A2, FBN1 and

**Fig. 2 Rearrangement of dermal fibroblasts and ECM via skin tension homoeostasis. a** Fluorescence images of ex vivo cultured skin maintaining (upper columns) or releasing (lower columns) tensional homoeostasis. Nuclei (left panels) and Type 1 collagen fibres (right panels) are shown. Scale bars, 20 μm. Orientation of nuclei distribution was plotted between 0 and 180 degrees (anterior–posterior; mean ± SD). SHG imaging (left) and FFT analysis (right) of the alignment of dermal collagen fibres in ex vivo cultured skins maintaining (upper) or releasing (lower) tensional homoeostasis. **b** Fluorescence images of Bell's model (upper columns), the THS model (middle columns) and the released model (lower columns). Nuclei (left panels) and SHG imaging of Type 1 collagen (right) are shown. Scale bars, 20 μm. Distribution plots show the dermal fibroblast nuclei orientation and Type 1 collagen fibre alignment. **c** Z-stack confocal imaging (left panels) and Imaris 3D rendering (right panels) of HSEs stained with a WGA-Alexa488 conjugate are shown. Scale bars, 20 μm. **d** Fluorescence images of HSEs stained with a WGA-Alexa488 conjugate (left panels), stained with a Phalloidin-Alexa594 conjugate (middle panels), and the merged images (right panels). Scale bars: 20 μm. **e** Schematic representation of cell shape and nuclear morphology through the rearrangement of cellular matrices by tensional homoeostasis.

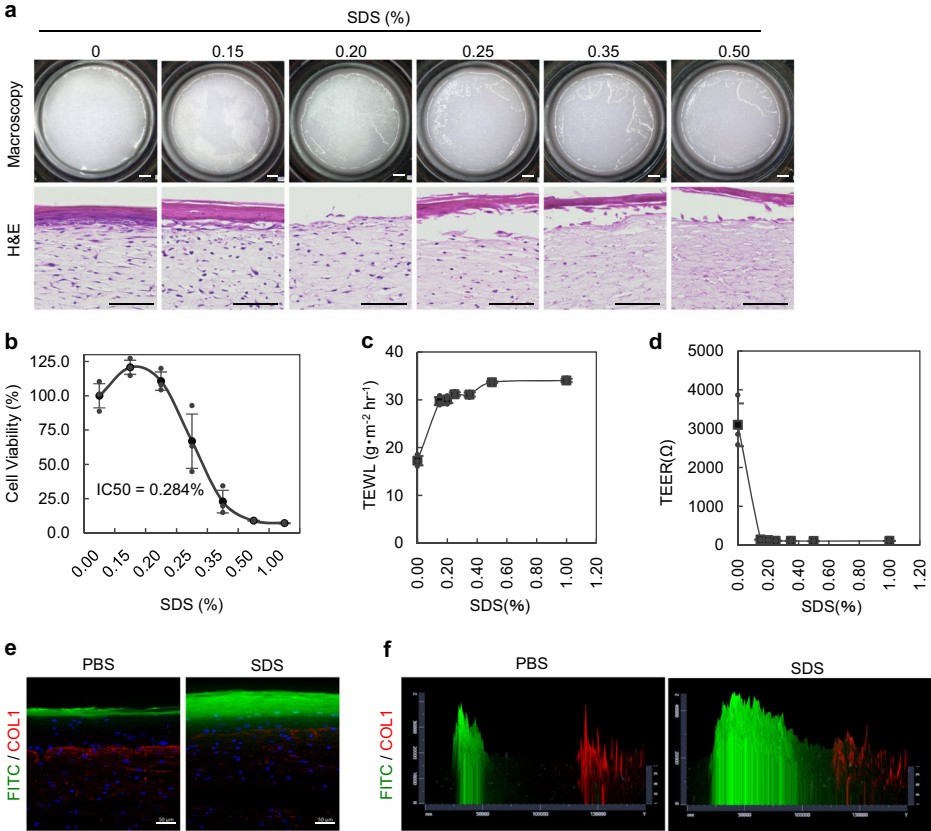

**Fig. 3 Analysis of THS model skin barrier function. a** Viability of cells in the THS model was detected by skin irritation tests using SDS solution (0, 0.15, 0.20, 0.25, 0.35 and 0.50%). Macroscopy images of epidermal repellency (upper columns) and histological analysis of cytotoxicity using H&E staining (lower columns). Scale bars, 1 mm in upper columns and 50 μm in lower columns. **b** MTT assays showed SDS-induced cytotoxicity in skin equivalents with half of the maximum inhibitory concentration (IC50): 0.284%. **c**, **d** TEWL (**c**) and TEER (**e**) analyses of the THS model with concentration-dependent stimulation with SDS. **e**, **f** Confocal images of vertical sections (**e**) and a 2.5D intensity plot (**f**) of the THS model topically treated with a fluorescein solution. To evaluate epidermal barrier function, these skin equivalents were pre-treated without (left panels) or with (right panels) 0.15% SDS solution. Scale bars, 50 μm.

ELN[44]. In the IAM, TGF-β treatment significantly induced the expression of genes, including these ECM-related genes (Fig. 4h). In immunohistochemical analyses of the THS model, we examined changes in genes related to epidermal morphology (KRT5, KRT10, Ki67 and CLDN1), cell proliferation (Ki67) and ECM production (COL1 and COL4) caused by the stimulation of ATRA, MAP and TGF-β in both administration models. ATRA or MAP administration stimulated protein expression, epidermal cell proliferation and successive morphological changes, including changes to thickness and maturation of the stratum corneum, which was observed in both models (Fig. 4i, j). In the dermal layer, COL1 fibre was induced according to the direction of traction force by all stimulants in both models (Fig. 4i, j). COL1 and COL4 matrix induction was clearly observed following

ATRA, MAP and TGF-β stimulation in the IAM (Fig. 4i). The same effectiveness of ATRA and MAP was observed in the TAM (Fig. 4j). These results indicate that reproduction of tension homoeostasis was effective in responsiveness to several drugs including ATRA, MAP and TGF-β.

**Reproduced tensional homoeostasis modulates mechanical stress signalling.** Tensional homoeostasis is known to regulate cell functions and fate determinations through mechanical stress signals[6,45]. We finally investigated the relationships between mechanical stress signalling and traction-force balance in these models. Integrin subunit alpha 2 (ITGA2), which adheres to collagen fibres and transduces mechanical stress signals into

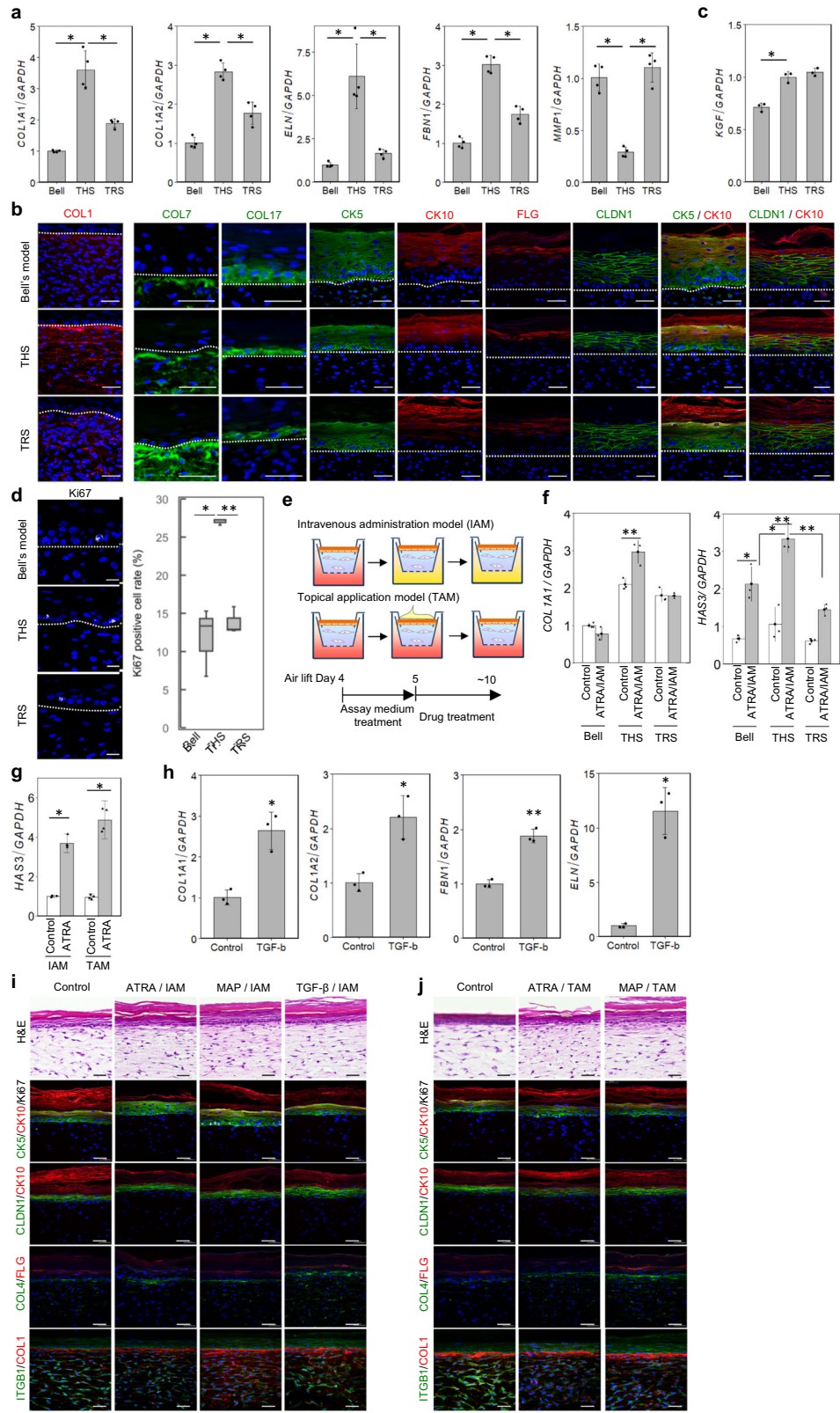

cells[46,47], was found to be induced in the THS model compared with its levels in the Bell's and TRS model (Fig. 5a). Mechanical stress-regulated integrin-associated signalling proteins such as FAK, mitogen-associated protein kinase (MAPK) and Rho A, which regulated mechanical feedback by the activation of ROCK; further, ROCK activated non-muscle myosin 2 contractility by phosphorylation of myosin light chain (MLP) and upregulated

factors in the mechano-transduction pathways such as *myocardin-related transcription factor A* (*MRTF-A*). The gene *MRTF-A* encodes a transcription coactivator that stimulates cellular contractility and fibroblast to myofibroblast differentiation in vitro and in vivo[18], and its expression was induced only in the THS model (Fig. 5a). Related to the activation of *MRTF-A*, the relative expression level of *ACTA2*, which is known as a

**Fig. 4 Analysis reveals that the THS model promoted function through tension homoeostasis. a, c** Real-time PCR analysis of dermal ECM proteins and epithelial–mesenchymal interaction-related genes (*COL1A1*, *COL1A2*, *FBN1*, *ELN*, *MMP1* and *KGF*) of HSEs. *$P < 0.001$, as assessed by Dunnett's test following two-way analysis of variance (ANOVA; $P < 0.001$); error bars represent the standard deviation. (**b**) Immunohistochemical analyses of epidermal and dermal morphogenesis markers in skin equivalents. These samples were stained with COL1, COL7, COL17, CK5, CK10, CLDN1, FLG and Ki67 antibodies. The dotted lines indicated boundary between epidermis and dermis. Scale bars, 50 μm. **d** Analysis of the Ki67-positive basal keratinocytes in skin equivalents. Immunohistochemical analyses (left) and Ki67-positive cell ratio (right) were shown. Scale bars, 20 μm. *$P < 0.01$ and **$P < 0.001$, as assessed by Dunnett's test following two-way analysis of variance (ANOVA; $P < 0.001$). **e** Schematic representation of the methods used for evaluating molecular activity. Functional molecules were applied to skin equivalents by dissolving in medium (as shown by IAM) or lotion (as shown by TAM) and treating for 6 h to 5 days. **f** The reactivity to ATRA in skin equivalents (Bell's model, THS model, and TRS model) was evaluated by mRNA expression levels of *HAS3* and *COL1A1* in the IAM assay model. *$P < 0.01$ and **$P < 0.001$, as assessed by Tukey–Kramer test following two-way analysis of variance (ANOVA; $P < 0.001$); error bars represent the standard deviation. **g** The reactivity to ATRA when using the IAM or TAM was evaluated by mRNA expression levels of *HAS3*. *$P < 0.001$, as assessed by Tukey–Kramer test following two-way analysis of variance (ANOVA; $P < 0.001$); error bars represent the standard deviation. **h** The reactivity to TGF-β in the THS model when using the IAM was evaluated by the mRNA expression levels of *COL1A1*, *COL1A2*, *FBN1* and *ELN*. *$P < 0.01$ and **$P < 0.001$, as assessed by two-tailed Student's *t*-tests; error bars represent the standard deviation. **i, j** Immunohistochemical analyses of skin equivalents for reactivity to ATRA, MAP and TGF-b when using IAM (**i**) or TAM (**j**). Scale bar, 50 μm.

myofibroblast marker, was also significantly upregulated only in the THS model. These results were confirmed by immunohisto-chemical analyses. ITGA2 and ITGB1 were observed in the plasma membrane of NHDFs in the dermal layer of the THS model. In contrast, ITGA2 expression was observed at a relatively low level in NHDFs in the Bell's and TRS models relative to that of the THS model (Fig. 5b). MRTF-A protein showed increased nuclear localization only in the THS model. The αSMA-positive myofibroblasts were localized beneath the basement membrane only in the THS model (Fig. 5b, c).

We next examined whether mechanical stress signalling is involved in tensional homoeostasis and whether it affects skin functions. The THS and TRS models were used, and a specific ROCK inhibitor, Y-27632, was found to significantly inhibit the tissue contraction induced by tension release (Fig. 5d, e). Y-27632 treatment also disrupted NHDF nuclei, the cytoskeleton (as observed by phalloidin staining), and cell membrane alignment (as shown by WGA staining) to a greater degree in the TRS model than it did in the THS model (Fig. 5f). Y-27632 treatment also affects the subcellular localization of MRTF-A and αSMA, which are downstream signalling molecules of ROCK (Fig. 5g, h). Y-27632 treatment of the THS model suppressed the distribution of collagen fibres (Fig. 5i). Although the gene expression of *COL1A1*, *COL1A2*, *FBN1* and *ELN* was inhibited in both the Y-27632-treated THS model and the TRS model (Fig. 5j), *MMP1* expression was drastically induced by Y-27632 treatment and the release of tensional homoeostasis (Fig. 5j). These results suggest that MRTF-A-mediated regulation of ROCK signalling is involved in the regulation of skin structure and function by tensional homoeostasis in the THS model, as summarized in Fig. 6.

## Discussion

In this study, the tensional homoeostasis plays important roles for skin tissue cell rearrangement through the expression of ECMs and skin functions including the responses to several chemicals and cytokines in the HSE model through mechanical stress sig-nalling. It is suggested that tensional homoeostasis is involved in maintaining skin homoeostasis including skin pathology and functions of natural skin. These findings contribute to under-standing skin homoeostasis and functions through tensional homoeostasis. THS model provides an animal experiment alter-native model that can be used for analyses of the molecular mechanisms of the skin physiological functions under tensional homoeostasis.

During embryogenesis, mechanical stimulations regulate complex morphogenic processes, including cell migration, dif-ferentiation and alignment of cells and tissues within organs, such

as the eyes, lung and brain[9,48,49]. In skin development, tissue tension is known to regulate the cell alignment of epidermal keratinocytes and dermal fibroblasts through the combination with ECM fibres and appendages such as hair follicles and align with skin tension direction, which is called Langar's cleavage line[12]. This tensional homoeostasis, which includes tension and mechanical stresses, is thought to contribute to physical proper-ties such as viscoelasticity, stiffness, and morphological char-acterization in features such as hair flow and skin texture[8,13,50]. During skin ageing, the morphology of dermal fibroblasts changes from a spindle shape to spherical shape, which is thought to be caused by the loss of tension transmission; the loss itself occurs through the degradation of several collagen fibres by proteases such as MMP1, 3 and 9[20,21,51]. In the current study, the tensional homoeostasis of living skin was revealed to be involved in maintaining the skin tissue structure. We found by histological and mathematical analyses that the tension mechanical stress contributed to regulating the cell alignment of dermal fibroblasts through the rearrangement of collagen fibres and contracted the skin structure through the reproduction of the horizontal traction force in the THS model, but not Bell's and TRS models. These findings suggest that tensional homoeostasis is an essential factor for reconstructing skin structure in vitro.

The skin plays a crucial role in protecting deep organs from extrinsic stress, such as dryness, chemical invasion, light irra-diation, thermal stress, and mechanical stress[10]. Current mechano-biological studies show that mechanical stimulations, such as external tension loading and changes in substrate phy-sical properties, play important roles in skin homoeostasis through the control of cell functions such as dermal cell pro-liferation and migration, and myofibroblast differentiation; an additional role is the regulation of metabolic factors, such as collagen, elastin and MMPs. These mechanical stimulations of skin are thought to be involved in physiological phenomena accompanying dynamic extrinsic stress, such as wound healing. Previous study reported that MRTF-A act as a promoter of retinoic acid-induced neural-like differentiation of adult bone marrow-derived mesenchymal stem cells[52]. It suggests that ten-sional homoeostasis is involved in drug responsibility including ATRA. However, the role of tensional homoeostasis in main-taining skin function remains unclear[19]. In the current study, we demonstrated that tensional homoeostasis induced in the HSE model and the THS model significantly increased the expression of ECM factors, including *collagen type 1 1alpha 1 chain*, *fibrillin-1*, *elastin* and *MMP1*. Furthermore, tensional homoeostasis sti-mulated not only dermal fibroblast functions but also epidermal keratinocyte proliferation and differentiation, suggesting the presence of epithelial–mesenchymal interactions through KGF

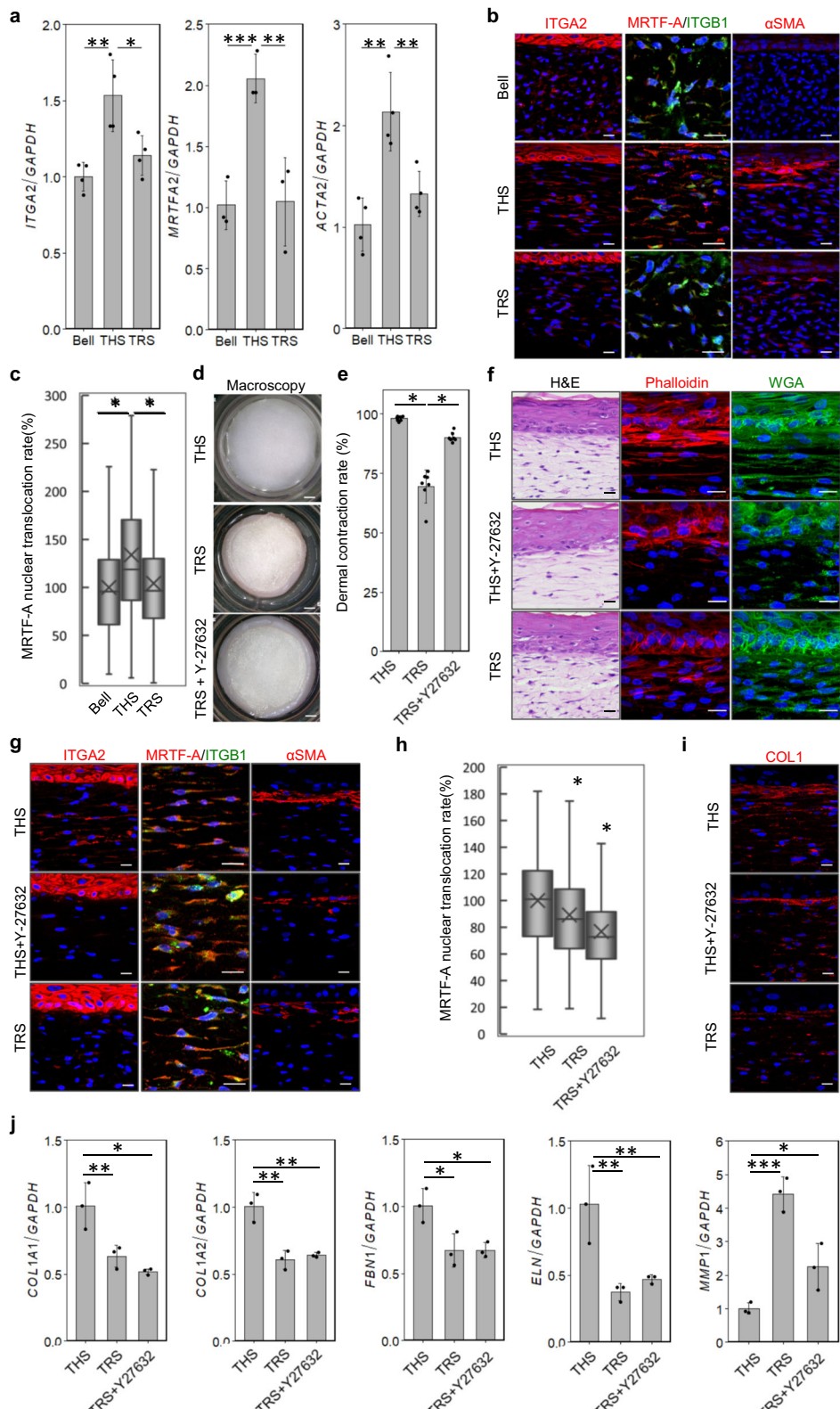

production. In the THS model, it was also enhanced the responsibilities of the gene expressions such as *collagen type 1 1alpha 1 chain* in dermal fibroblasts and *HAS3* in keratinocytes for functional molecules such as ATRA and MAP. These findings suggest that tensional homoeostasis into the dermis is an essential factor in skin functions in both keratinocytes and dermal fibroblasts.

Mechanical stresses are well known to be transmitted to cells through the plasma membrane and cytoskeleton, which are regulated by biological signalling through protein-folding changes, localization and interaction with mechano-transduction-related proteins such as integrins, YAP/TAZ and MRTF-A[6,45,53]. It was reported that actomyosin-mediated tissue tension contributes to the 3D body shape because a mutation that disrupts

**Fig. 5 Tension homoeostasis regulates skin morphogenesis. a** Real-time PCR analysis of mechanical stress-related genes (*ITGA2, MRTF-A* and *ACTA2*) in HSEs. *P < 0.05, **P < 0.01 and ***P < 0.001, as assessed by Dunnett's test following two-way analysis of variance (ANOVA; P < 0.001); error bars represent the standard deviation. **b** Immunohistochemical analyses of mechano-sensitive proteins in HSE. These samples were stained with anti-ITGA2, anti-ITGB1, anti-MRTF-A and anti-αSMA antibodies. Scale bars, 20 μm. **c** Calculation of the nuclear localization of MRTF-A in cells (**b**). *P < 0.05, as assessed by Dunnett's test following two-way analysis of variance (ANOVA; P < 0.001); error bars represent the standard deviation. **d** Macroscopic images of the THS model (upper) and TRS model without (middle) or with (lower) Y-27632 treatment. Scale bars; 1 mm. **e** Calculation of the shrinkage rate of (**d**). *P < 0.001, as assessed by Tukey–Kramer test following two-way analysis of variance (ANOVA; P < 0.001); error bars represent the standard deviation. **f** Histological analysis of the THS model without (upper columns) or with (middle columns) Y-27632 treatment; the TRS model is also shown (lower columns). H&E staining (left), phalloidin staining (middle) and WGA staining are shown. Scale bars, 20 μm. **g** Immunohistochemical analyses of focal adhesion proteins and mechano-transducer proteins in the THS model without (upper columns) or with (middle columns) Y-27632 treatment; the TRS model (lower columns) is also shown. Scale bars, 20 μm. **h** Calculation of the nuclear localization of MRTF-A in cells (**g**). *P < 0.05, as assessed by Dunnett's test following two-way analysis of variance (ANOVA; P < 0.001); error bars represent the standard deviation. **i** Immunohistochemical analyses of collagen fibres in the THS model without (upper) or with (middle) Y-27632 treatment; the TRS model (lower). Scale bars, 20 μm. **j** Real-time PCR analysis of dermal ECM-related genes in HSEs. *P < 0.05, **P < 0.01 and ***P < 0.001, as assessed by Dunnett's test following two-way analysis of variance (ANOVA; P < 0.001); error bars represent the standard deviation.

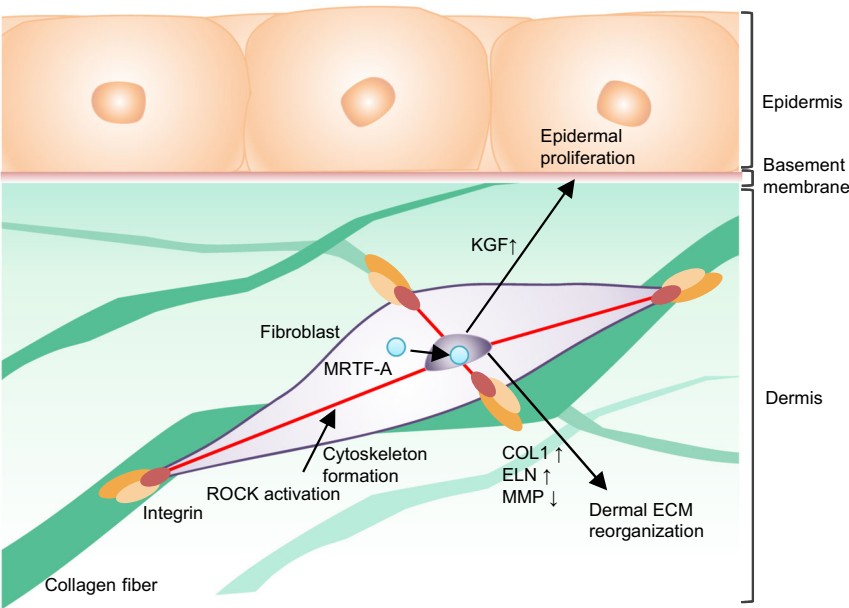

**Fig. 6 Schematic representation of the regulation of skin structure and function by tensional homoeostasis.** ROCK signalling maintains cytoskeleton formation and promotes the nuclear localization of MRTF-A, which induces the gene expression of ECM factors and KGF.

YAP leads to abnormal morphogenesis and tissue misalignment[1]. In the skin, the mechano-sensation of tissue stiffness controlled by integrins activates the Rho/ROCK pathway, which regulates tissue stiffness via collagen synthesis, keratinocyte proliferation, epidermal hyperplasia and tumour growth through β-catenin stabilization, nuclear accumulation and transcriptional activation[7]. It is also thought that the disruption of tensional homoeostasis and abnormalities in the mechano-transduction pathways are involved in ageing skin dysfunctions, such as tension deficiency, hypertrophic scarring and keloid-associated hypertension[19,21]. In this study, the tensional homoeostasis in the HSE models activated the mechano-transduction pathway, as evidenced by both gene and protein expression of integrin α2 and the nuclear localization of MRTF-A (which was mediated by ROCK signalling); further, the inhibitor Y-27632 significantly reduced the constriction of the artificial skin in the THS model. We also found that reproduction of tensional homoeostasis induced myofibroblast differentiation, which indicated that F-actin and collagen fibre formation aligned with the tension direction; these results suggest the activation of epithelial–mesenchymal interactions by KGF. These results indicate that tensional homoeostasis plays essential roles in skin tissue formation and functions in skin homoeostasis through the regulation of the mechano-transduction pathway.

In conclusion, we demonstrated that tensional homoeostasis is crucial for skin structures and functions both in vivo and in vitro through mechano-transduction signalling. Our developed a traction-force-induced HSE model, which will contribute to analyze the molecular mechanisms of skin physiological functions under the tensional homoeostasis as an alternative to animal experiments. Further studies on the molecular mechanisms of tensional homoeostasis, the mechano-transduction pathway and the advanced reproduction of mechanical properties of dermis by introduction of other ECM such as elastic fibre and proteoglycans will provide new insight into skin homoeostasis in the basic science, healthcare and pharmaceutical fields. In the future, it is expected that a next-generation skin organ system model will be developed that exhibits various skin functions and structures such as tensional homoeostasis, skin appendages and texture.

## Methods

**Animals.** C57BL/6NCrSlc mice (7–9 weeks old, female) were purchased from Japan SLC, Inc. (Shizuoka, Japan). Animal care and handling were performed in accordance with the National Institutes of Health guidelines and was approved

Institutional Animal Care and Use Committee at RIKEN Kobe Branch (permit no. A2014-02-14).

**Cell culture**. Normal human epidermal keratinocytes (NHEKs) were purchased from KURABO Industries LTD (Osaka, Japan). The NHEKs were maintained in HuMedia-KG2 (KURABO). NHEKs were seeded into 150 mm dishes and grown in a humidified atmosphere of 5% $CO_2$ at 37 °C.

Normal human dermal fibroblasts (NHDFs) were purchased from KURABO and were maintained in Dulbecco's modified Eagle's medium (FUJIFILM Wako Pure Chemical Corporation, Osaka, JAPAN) that was supplemented with 10% foetal bovine serum (Biowest, Rue du Vieux Bour, Nuaillé, France), 100 units mL$^{-1}$ penicillin, 100 µg mL$^{-1}$ streptomycin (Thermo Fisher Scientific, MA, USA) and 10 ng ml$^{-1}$ bFGF (PeproTech, NJ, USA). NHDFs were seeded in 150 mm dishes and grown in a humidified atmosphere of 5% $CO_2$ at 37 °C.

**Reconstruction of the THS model, Bell's model and tension-released skin (TRS) model**. The THS model was generated by the following procedure. A total of $9 \times 10^5$ NHDF cells were resuspended in 1.8 ml of 4 mg ml$^{-1}$ bovine dermis derived native collagen solution (KOKEN CO. Ltd., Tokyo, Japan) that was mixed with 1× DMEM (Thermo Fisher Scientific), 10 mM HEPES (Dojindo, Kumamoto, Japan), 10 mM NaHCO$_3$ (FUJIFILM Wako Pure Chemical Corporation), 1% penicillin/streptomycin and 5% FBS; then the cells were plated in a high-density translucent membrane cell culture insert (0.4 µm pore size) (Corning, NY, USA) in 6-well plates. After solidification of the collagen solution, 0.6 ml of collagen solution mixture containing $3.7 \times 10^6$ NHDF cells was plated on top of the first layer of fibroblasts. Dermal equivalents were held in place by a snapwell culture insert (Corning). NHEKs were seeded at a density of $1.1 \times 10^6$ cells on fixed dermal equivalents, and they were cultured under submersion conditions for 4 days in a humidified atmosphere at 37 °C with 5% $CO_2$ and 12.5% $O_2$. After maintenance in a submersion culture, skin equivalents were exposed to the air–liquid interface in a humidified atmosphere at 37 °C with 5% $CO_2$. The medium for culturing skin equivalents consisted of DMEM supplemented with 10% FBS, 1% penicillin/streptomycin, 5 µg ml$^{-1}$ insulin (FUJIFILM Wako Pure Chemical Corporation), 1 mM magnesium ascorbyl phosphate (MAP, FUJIFILM Wako Pure Chemical Corporation), 10 ng ml$^{-1}$ bFGF and 1 µM hydrocortisone (FUJIFILM Wako Pure Chemical Corporation), was replaced with fresh medium every 48 to 72 h.

Bell's model was generated by maintaining contracted dermal equivalents in a floating culture for 7 days. The TRS model was generated by resection of the THS model from the culture insert after 4 days of culture in air lift conditions.

**Ex vivo culture of mouse dorsal skin**. A snapwell was attached onto shaved mouse dorsal skin with cyanoacrylate adhesives, and the surrounding skin was excised with scissors and cultured in HSE culture medium for 24 h. In the release group, skin was excised with scissors from the area where the snapwell was located and then was cultured.

**H&E staining, immunohistochemistry**. For histological analysis, skin equivalents from three independent experiments were fixed within 4% formaldehyde and were embedded in paraffin or Tissue-Tek O.C.T. Compound (Sakura Finetek Japan Co., Ltd., Tokyo, Japan). Haematoxylin-eosin staining was performed on paraffin sections (10-µm thick). The stained sections were observed using Axio Scan.Z1 (Carl Zeiss, Oberkochen, Germany). For fluorescent immunohistochemistry, frozen sections (10 and 50 µm) and paraffin sections (10 µm) were prepared and immunostained as previously described[29,31]. Details of primary and secondary antibodies and associated epitope recovery methods are included in Table 1. All fluorescence microscopy images were captured with an LSM780 confocal microscope (Carl Zeiss).

**Multiphoton setup**. SHG imaging was performed as described previously[54]. Briefly, multiphoton imaging was performed using an LSM780 confocal microscope system (Carl Zeiss) with a ×25, 0.8-NA objective lens (LD LCI Plan-Apochromat 25×/0.8 Imm Korr DIC M27; Carl Zeiss). We typically recorded three $340 \times 340 \times 30$ µm z-stacks in every sample, with a 1 µm z-step and 0.332 µm pixel size. No photodamage was observed under these experimental conditions.

**Quantitative analysis of epidermal cell proliferation**. The proliferation rate of epidermal cells was quantified by counting keratinocytes with Ki67-positive nuclei located at the basal or immediately supra-basal epidermis. The measurement area was a total of 9 images obtained by randomly acquiring three z-stack images of $340 \times 340 \times 11$ µm from HSE tissues that were derived from three different wells.

**Quantitative analysis of cell nuclei orientation**. Dermal nuclei orientation was calculated by outlining the nuclear counterstain images and placing an ellipse in relation to individual segmented nuclei using FIJI. The orientation of each nucleus was defined by the angle at the major axis of the ellipse[12].

**Quantification of F-actin, cytoplasm and collagen fibre alignment**. Anisotropic alignment of dermal actin, the plasma membrane, and collagen fibres was assessed using two-dimensional fast Fourier transform (2D-FFT) analysis, as previously described[12]. Briefly, uncompressed images of dermal actin labelled with phalloidin, plasma membranes labelled with WGA, and collagen fibres detected by SHG imaging were analysed with the FFT function of ImageJ, and the radial summation of the FFT frequency plot was calculated using the Oval profile plug-in. The degree of fibre alignment was reflected by the shape and height of the major peak in the FFT alignment plot. Images with oriented actin fibres resulted in a prominent peak

**Table 1 Primary and secondary antibodies.**

| Antibodies | Source | Cat# | Dilution ratio |
|---|---|---|---|
| *Primary* | | | |
| Anti-alpha smooth muscle Actin antibody | Abcam | Ab5694 | 1/100 |
| Anti-c-Jun antibody [E254] | Abcam | Ab32137 | 1/250 |
| Anti-Cytokeratin 10 antibody | Abcam | Ab9026 | 1/200 |
| Anti-Cytokeratin 5 antibody | Abcam | Ab52635 | 1/200 |
| Claudin 1 antibody | Thermo Fisher Scientific | 71-7800 | 1/100 |
| Anti-Collagen I antibody | Abcam | ab34710 | 1/200 |
| Anti-Collagen IV antibody | Abcam | ab6586 | 1/500 |
| Anti-Filaggrin antibody [SPM181] | Abcam | ab17808 | 1/50 |
| Anti-Integrin alpha 2 antibody [EPR5788] | Abcam | ab133557 | 1/250 |
| Anti-Integrin beta 1 antibody | Abcam | ab30388 | 1/100 |
| Anti-Ki67 antibody | Abcam | ab156956 | 1/150 |
| Anti-Collagen III antibody | Abcam | ab6310 | 1/500 |
| Anti-Collagen VII antibody | Abcam | ab6312 | 1/1000 |
| Anti-Collagen XVII antibody | Abcam | ab184996 | 1/100 |
| Phalloidin-Alexa594 conjugate | Thermo Fisher Scientific | R415 | 1/1000 |
| Alexa Fluor™ 488 Phalloidin | Thermo Fisher Scientific | A12379 | 1/1000 |
| Anti-Mkl1/MRTF-A antibody | Abcam | ab113264 | 1/100 |
| Wheat Germ Agglutinin, Alexa Fluor™ 488 Conjugate | Thermo Fisher Scientific | W11261 | 1/400 |
| *Secondary* | | | |
| Alexa Fluor 594 donkey anti-rabbit IgG | Thermo Fisher Scientific | A21207 | 1/500 |
| Alexa Fluor® 594 donkey anti-mouse IgG (H + L) antibody | Thermo Fisher Scientific | A21203 | 1/500 |
| Alexa Fluor® 488 donkey anti-rabbit IgG (H + L) | Thermo Fisher Scientific | A21206 | 1/500 |
| Alexa Fluor® 488 donkey anti-mouse IgG (H + L) | Thermo Fisher Scientific | A21202 | 1/500 |
| Alexa Fluor 633 goat anti-rat IgG (H + L) | Thermo Fisher Scientific | A21094 | 1/500 |

The table shows the commercially available antibodies which were used for immunohistochemical analyses.

**Table 2 Primers for real-time PCR.**

| Gene | Forward | Reverse |
|------|---------|---------|
| *SYBR* | | |
| GAPDH | TCTGACTTCAACAGCGACAC | CCCTGTTGCTGTAGCCAAATTC |
| KRT5 | AATGCAGACTCAGTGGAGAAGG | ACTGCCACTGCCATATCCAG |
| KRT6 | GGAGCTGATGAATGTCAAGCTG | GCCATAGCCACTGGAGACG |
| COL1A1 | AGGAAGGCCAAGTCGAGG | CCGGGGCAGTTCTTGGTC |
| COL1A2 | GATGGTGAAGATGGTCCCAC | GTCCAGGGCCAAGTCCAAC |
| ELN | TCCTGCTGTCCATCCTCCAC | AAAAGACTCCTCCAGGAACTCC |
| MMP1 | AGAATGTGCTACACGGATACCC | CCAGTGTTTTCCTCAGAAAGAGC |
| c-JUN | TTGCGGCCCCGAAACTTGTG | AGGCGTTGAGGGCATCGTCATAG |
| ITGA2 | TGTCCTGTTGACCTATCCACTGC | TGTGAGAAAACCTCCAGTTCCC |
| MRTF-A | ATGCCGCCTTTGAAAAGTCCAG | AGCCGAGGTCTCTTCCAAAATG |
| TGFb1 | AAGGACCTCGGCTGGAAGTG | CGCCCGGGTTATGCTGGTTG |
| HAS3 | CGCAGCAACTTCCATGAGG | AGTCGCACACCTGGATGTAGT |
| KGF | AACACAGTGGTACCTGAGGATCG | ACTTTCCACCCCTTTGATTGCC |
| *TaqMan* | | |
| Gene | Source | Cat# |
| GAPDH | Thermo Fisher Scientific | Hs02786624_g1 |
| FLG | Thermo Fisher Scientific | Hs00856927_g1 |
| ACTA2 | Thermo Fisher Scientific | Hs00426835_g1 |
| FBN1 | Thermo Fisher Scientific | Hs00171191_m1 |

The table shows the primer sequence that was used for real-time PCR analyses.

centred at the principal axis of fibre alignment, whereas images with unaligned actin fibres resulted in an alignment plot with a broad peak or no peak.

**Evaluation of epidermal barrier function by TEER measurement**. Transepithelial electrical resistance (TEER) measurements were performed using EndOhm (World Precision Instruments, Sarasota, FL, USA). Prior to the measurement, the skin equivalent was incubated inside a biosafety cabinet for 15 min to allow equilibration to ambient temperature and humidity. After washing with D-PBS (−), the skin equivalent was inserted into the EndOhm instrument. Values were recorded beginning at 10 s.

**Evaluation of epidermal barrier function of HSEs by TEWL measurement**. Transepidermal water loss (TEWL) was measured in the HSEs ($n = 3$) using a VAPO SCAN system (ASAHI BIOMED, Kanagawa, Japan) according to the instruction manual. Prior to collecting the measurement, the skin equivalent was incubated inside the biosafety cabinet for 15 min to allow equilibration to the ambient temperature and humidity of the room. The measurement was repeated three times, and the average and standard deviation of each well were calculated.

**Evaluation of epidermal barrier function by skin irritation molecule application**. The epidermal barrier function of the HSEs was analysed according to the method of the Organisation for Economic Co-operation and Development (OECD) test guidelines 431 and 439. Various concentrations of SDS solution (0.15, 0.20, 0.25, 0.35 and 0.50%) were applied to the surface of the skin equivalent, and the epidermis was exposed for 60 min to a humidified atmosphere of 5% $CO_2$ at 37 °C. D-PBS (−) was used as a negative control. At the end of the exposure, the surface of the epidermis was gently washed twice with D-PBS (−). As a posttreatment incubation period, the skin equivalent was cultured in maintenance medium for 24 h. Then, MTT was added to the maintenance medium at a concentration of 0.5 mg mL$^{-1}$, and the epidermis was cultured for an additional 2 h to allow MTT to be incorporated into the skin equivalent. After that, we immersed the skin equivalent in isopropyl alcohol for more than 16 h to extract the precipitated formazan product. Cell viability was calculated by measuring the absorbance at 570 nm to determine the concentration of the formazan product. All tests were performed in triplicate.

**FITC diffusion analysis**. HSEs were topically administered a 0 or 0.15% SDS solution for 15 min and then were washed with PBS (−) at room temperature. All HSEs were cultured for 24 h and then were administered a 5 mM FITC/PBS solution for 1 h at 37 °C. After FITC removal, they were cultured for 24 h and fixed with 4% paraformaldehyde. The FITC diffusion pattern in the epidermis was determined with an LSM 880 confocal microscope, which was analysed by ZEN software (Carl Zeiss).

**Evaluation of HSE responsiveness for functional molecules**. All HSEs were pretreated with test medium (DMEM supplemented with 1% FBS, 1% penicillin/streptomycin, 5 µg mL$^{-1}$ insulin and 1 µM hydrocortisone) for 48 h before drug

application. All functional molecules were administered to HSEs in an air–liquid interface culture beginning on day 5 for 6 h and lasting for 5 days. Testing was performed on two drug administration models, an intravenous administration model (IAM) or a topical application model (TAM). In IAM, ATRA (Merck KgaA, Darmstadt, Germany), MAP and TGF-β (Pepro Tech, NJ, USA) were dissolved in test medium to final concentrations of 10 µM, 1000 µM and 10 ng mL$^{-1}$, respectively. In TAM, ATRA and MAP were dissolved in a test lotion formulation (0.2% hydroxypropyl cellulose dissolved in deionized water) to final concentrations of 10 and 1000 µM, respectively. HSE was topically applied in 200 µL of a lotion sample for 60 min, and then the sample was aspirated and incubated for 6–72 h.

**Real-time PCR**. HSEs ($n = 4$) were homogenized with zirconium oxide beads and stainless steel beads (Bertin Technologies, Montigny-le-Bretonneux, France). RNA was isolated using phenol/chloroform extraction and a Rneasy Plus Mini kit (Qiagen, Venlo, Netherlands), and it was reverse transcribed using SuperScript VILO (Thermo Fisher Scientific) to generate cDNA. Real-time PCR was performed on an Applied Biosystems QuantStudio 12K Flex (Thermo Fisher Scientific) using a SYBR Premix Ex Taq II (TaKaRa Bio, Shiga, Japan) or a TaqMan Gene Expression Assay system (Thermo Fisher Scientific). The expression level of each target gene was normalized to GAPDH expression. The primer pairs used for real-time PCR are listed in Table 2.

**Quantification of the MRTF-A nuclear translocation rate**. MRTF-A protein localization in each HSE, including THS, TRS, Bell's model and the THS model treated for 72 h with 10 nM Y-27632, was visualized by immunohistochemical staining. The MRTF-A nuclear translocation rate was calculated by outlining the nuclear-counterstained images and measuring the mean intensity of MRTF-A signals in the nuclear area with FIJI in over 240 nuclei.

**Statistics and reproducibility**. Microsoft Excel (Microsoft, Redmond, WA, USA) and was used for statistical analysis using the student's $t$ test. All values were expressed as means ± standard deviation (SD). Analysis of samples was performed at least in triplicate and averaged. The difference between groups was regarded considerable at $P < 0.05$. All experiments were repeated at least three times.

**Reporting summary**. Further information on research design is available in the Nature Research Life Sciences Reporting Summary linked to this article.

## Data availability

The source data for the figures presented in this study have been included as a source data file. All data are available from the corresponding authors upon reasonable request.

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

## Acknowledgements

We thank lab members in RIKEN BDR, especially Dr. A. Noma, Ms. C. Fukuda, Ms. T. Iga, Ms. Y. Morioka and Ms. M. Takase, the animal facilities of BDR and Dr. H. Shichiri, Mr. K. Sakamoto and Dr. K. Tezuka of Organ Technologies Inc. for technical assistance. We also thank Dr. K. Takagi, and Mr. Y. Honma of Rohto Pharmaceutical Co., Ltd., and Mr. A. Inoue and Dr. T. Mikuniya of Meiji Seika Pharma Co., Ltd. for their support and encouragement on this project. This work was supported by JSPS KAKENHI (Grant number: JP16H01851 to T.T.). We would like to thank Rohto

Pharmaceutical Co., Ltd., Meiji Seika Pharma Co., Ltd. and Organ Technologies Inc. for their funding support.

## Author contributions

T.T., S.K. and M. Ogawa designed the research plans; S.K., A.T., M. Ogawa, M. Ono, N.S. and K.S. performed the experiments; S.K., A.T., M. Ogawa, M.T. and T.T. discussed the results; S.K., M. Ogawa. and T.T. wrote the manuscript.

## Competing interests

T.T. is a supreme technical advisor at Organ Technologies Inc. This work was partially performed under the condition of an Invention Agreement between RIKEN, Rohto Pharmaceutical Co., Ltd., Meiji Seika Pharma Co., Ltd. and Organ Technologies Inc. The remaining authors declare no competing interests.
