## [Peer Review File · Communications Biology]

Reviewers' comments:

Reviewer #1 (Remarks to the Author):

The article presents a refined human skin equivalent (HSE) model based on producing and maintaining HSEs under tensional homeostasis. The histology, epidermal differentiation, dermal ECM deposition and arrangement are analyzed and compared to HSEs assembled without tensional homeostasis or after tension release. The applicability of these models for preclinical drug testing is then assessed.

There is a real need for HSE models with closer resemblance to human physiology or pathophysiology for drug testing. The work is therefore very relevant. Overall the presented data are clear and the figures are nicely composed. The width of data presented and techniques used, allowing for detailed histological to functional assessment, is impressive.

1. My main specific concern is that few data are presented showing that the developed model would be more accurate for drug assessment than e.g. Bell's model. It is clear that the presented HSE model under tensional homeostasis in some aspects closer to human skin than what Bell's model is. However, that this translates into improved drug testing is not unequivocally shown.
2. Tension is also a promoter of fibrosis. How was the tension set so that it would provide the right tension to promote development of physiological but not pathophysiological fibrotic skin?

Specific comments on figures and data.

2. Figure 1a and b. This is done on murine skin. Mouse skin is softer than human skin; it would be more relevant to show this for human skin. However, I acknowledge that it may be challenging to obtain suitable human skin samples.
3. Figure 1f. In the human skin sample, papillary and reticular dermis can be clearly distinguished. This is not the case for the tension homeostatic (THS) model. It should be better analyzed if papillary and reticular dermis are established in the model. The collagen III staining (Figure 1g) could suggest that at least partial such division occurs. However, to me the antibody does not really produce the expected collagen III staining in skin, it appears more like a collagen IV staining. I could not find the antibody in the list, table 1.
4. Assessment of important anchoring structures of the skin, such as anchoring fibrils is missing.
5. Figure 1g, figure 3, figure 4g-j should be expanded to also include data from at least Bell's HSE.
6. Figure 1g, the cells on THS appear in general much larger than those in normal human skin.
7. Figure 4f. For other HSE models it has been shown that there is a dose response to all-trans retinoic acid in terms of collagen synthesis – lower doses (250 – 500 nM) stimulated collagen production; higher doses inhibited it (Deshpande et al., Arch Dermatol Res. 2014). The concentration used here 10 micromolar is high. Was the response to different doses tested?

Reviewer #2 (Remarks to the Author):

Very comprehensive and excellently written manuscript.

My major concern is that the authors over-emphasize the novelty of their findings. Most of these aspects have been demonstrated before, such as the suitability of skin models for skin irritation

testing, that they can be used to assess the skin barrier function and to assess drug-related effects. Maybe it would be better to focus on the novel aspects (model to study tensional hemostasis). It would be helpful to include a bit more information about the differences between the models that have been used in this study.

Additional comments:

Fig. 4b: The Ki67 staining seems to be very unspecific. The IF images are not convincing.

Fig. 4g (also Fig. 5a): The changes in the gene expression levels are minor – this holds for several of the qPCR data.

Fig. 4i: The control model (and also the other models) look pretty differentiated. Only very few, undifferentiated epidermal cells seem to be left. Overall, I don't see tremendous differences between the differently treated models. I feel it will be difficult to assess drug-related effects with that.

Fig. 5c: I doubt that there is a statistically significant difference given the huge standard deviations and that the error bars are overlapping.

Page 14, line 217-19: I can't conclude this from the figures.

Page 14, line 224-226: The Ki67 staining appears to be highly unspecific. Can't draw this conclusion from the provided images

Reviewer #3 (Remarks to the Author):

This manuscript reports on a tension homeostatic skin model that reproduces traction-force balance in the lateral direction. This model has a great potential for applications in drug screening and understanding the molecular mechanisms of drug effects, skin ageing and diseases. The authors have performed a series of detailed in vitro and in vivo studies to validate their model. To my best knowledge, the idea presented herein is novel and has significant implications in the area of drug discovery and skin disease modelling. Overall, the manuscript is well-written and easy to follow, although I have a few comments that may improve the quality of the manuscript.

1-Although the authors have touched on a few existing skin tissue equivalents in the introduction, their survey of literature is only limited on traditional transwell methods. Recently, a myriad of tissue engineering approaches have been used to recreate the skin tissues. Authors are encouraged to cite a few of recent papers that used microfluidics and 3D bioprinting to create bioengineered skin tissues.

2-In line 116, the authors mention that Bell's model has been traditionally used for creating contracted dermal equivalents. However, they do not mention why there is a need for a new model, such as the one presented in this work. I encourage the authors to expand on the shortcomings of the Bell's method further and describe how their approach overcomes these shortcomings.

3-What type and concentration of collagen was used to create the THS and TRS models? Given that the skin ECM consists of many other proteins including elastin, the authors are encouraged to discuss the rationale behind using only collagen in their experiments.

4-The authors used mice skin tissues to generate the ex vivo models (Figure 1b). Why mouse tissue was used in this work? Skin contraction in mouse and human is different.

Response to Reviewer #1

We have studied your comments carefully and found that you understood the value and significance of our study in this field. We are grateful for your evaluation and valuable suggestions for our manuscript. Our specific responses are listed below:

Comment 1: My main specific concern is that few data are presented showing that the developed model would be more accurate for drug assessment than e.g. Bell's model. It is clear that the presented HSE model under tensional homeostasis in some aspect closer to human skin than what Bell's model is. However, that this translates into improved drug testing is not unequivocally shown.

Our Response: We are grateful for your careful evaluation of our manuscript. The THS model and the TRS model will contribute to analyze skin physiological functions and pharmacological actions regulated by tensional homeostasis as an alternative to animal experiments. According to your comments, we have revised that our description is suitable to understand the applicability for drug assessment compared to Bell's model under tensional homeostasis in line 35-37, 101, 305-310, and 377-380.

Comment 2: Tension is also a promoter of fibrosis. How was the tension set so that it would provide the right tension to promote development of physiological but not pathophysiological fibrotic skin?

Our Response: The balance among extracellular forces exerted on cells by the ECM or neighbouring cells and the traction forces generated by cells themselves is termed tensional homeostasis (ref 5). Under tensional homeostasis, tissue maintain a constant morphology without contraction. Therefore, we prepared normal human dermal fibroblasts in collagen gel at a density close to that of natural dermis (Fig. 1f) and clamped the dermis of THS model to prevent contraction. In the THS model, spontaneous traction force by fibroblasts and the traction force from the culture insert balanced according to the law of action and reaction, and tensional homeostasis of the natural skin is reproduced.

As pointed out by your criticisms, the extrinsic tension would be a possibility to promote fibrosis (Harn HI *et al.*, Exp Dermatol. 2019). In addition, it has been reported that the traction force and proliferation activity of fibrosis-derived dermal fibroblasts is stronger than that of normal fibroblasts (Garner WL *et al.*, Wound Repair

Regen. 1995). Our THS model was constructed using normal human dermal fibroblasts without extrinsic mechanical tension loading. Immunohistochemical analysis showed that dermal fibroblasts were almost completely Ki67-negative, indicating no pathological fibroblast differentiation (Fig. 4d). For the above reasons, we consider that the THS model mimic physiological but not pathophysiological fibrotic skin environment.

Specific comment 1: Figure 1a and b. This is done on murine skin. Mouse skin is softer than human skin; it would be more relevant to show this for human skin. However, I acknowledge that it may be challenging to obtain suitable human skin samples.

Our Response: We are grateful that you have recognized that it is challenging to obtain suitable human skin samples. In this experiment, the intact human skin should be fixed with a glass ring and excised in order to discuss tension balance in the physiological environment. We also think that human skin is ideal for more precise discussion, because the biophysical properties of mouse and human skin are different. However, as pointed by you, it was difficult to obtain suitable human skin for ethical reasons by basic scientists. We believe that there is a certain similarity between the mouse and human regarding the tensional homeostasis in the skin. We will study on your pointed subject with clinicians in the further studies.

Specific comment 2. Figure 1f. In the human skin sample, papillary and reticular dermis can be clearly distinguished. This is not the case for the tension homeostatic (THS) model. It should be better analyzed if papillary and reticular dermis are established in the model. The collagen III staining (Figure 1g) could suggest that at least partial such division occurs. However, to me the antibody does not really produce the expected collagen III staining in skin, it appears more like a collagen IV staining. I could not find the antibody in the list, table 1.

Our Response: The papillary and reticulated dermis are highly distinctive structures in human skin. We think that the THS model has not yet been able to reproduce the structural features of the papillary and reticular dermis completely. In *in vitro* reconstruction of papillary and reticular dermis may require 3D rearrangement of fibroblast multi-lineages. We would like to perform this assay in a future study.

Following your suggestion, we have analyzed carefully and improved histogram adjustment of IHC images of type 3 collagen (Fig. 1g, and line 135).

Immunohistochemical analyses revealed that in addition to Type 1 and 3 collagen deposits throughout the dermis, the localization was clearly detected in the dermis near the basement membrane of the THS model. Similar collagen localization pattern, which is thought to be stimulated by epithelial and mesenchymal interactions, has also been reported in existing HSEs (Osborne R, *et al.* J Drugs Dermatol. 2009). We also added the antibody of type 3 collagen in the list, table 1.

Specific comment 3. Assessment of important anchoring structures of the skin, such as anchoring fibrils is missing.

Our Response: We are grateful for your suggestion to strengthen for our study. According to your suggestion, we have performed the immunohistochemical analyses of Type 7 collagen and Type 17 collagen in human skin, THS, TRS, and Bell's model (Fig. 1g and h, Fig. 4b) and added in the revised manuscript (line 138-141, and 215-219). We confirmed that the localization patterns of constituent proteins of hemidesmosomes and anchoring fibrils in basement membrane zone were consistent with human skin.

Specific comment 4. Figure 1g, figure 3, figure 4g-j should be expanded to also include data from at least Bell's HSE.

Our Response: In Fig. 1, we have demonstrated the construction of a novel THS models. Because, past many studies reported the immunohistochemical properties in Bell's model. Thus, we have analyzed of the expression patterns of wide variety of molecule in the immunohistochemical analyses of THS, TRS, and Bell's models in Fig. 4b.

In the Fig. 3, we have only demonstrated that the THS model is also applicable to the previously reported barrier function as a HSE model. In accordance with your comments, we have revised the description of the applicability of the THS model for safety testing (Line 35-37, 305-310, 377-380). The effects of tensional homeostasis on the response to skin irritants should be carefully discussed from the perspective of pharmacology and safety assessment, and we recognize that this is an important issue in the future study.

In the Fig. 4g-j, we have also performed immunohistological analysis on Bell's and TRS models, the staining pattern difference only slightly. It is well known that the culture period of the HSE models is limited *in vitro*. Thus, we have analyzed gene expressions but not protein accumulations of several molecules, because the current

culture period of THS model is not sufficient to detect the difference of protein accumulation in drug responsiveness. Thus, we will also develop a additional novel THS model for long-term culture analysis in a future study.

Specific comment 5. Figure 1g, the cells on THS appear in general much larger than those in normal human skin.

Our Response: In accordance with your comment, we reconfirmed the image. Because the THS model was sliced parallel to the tension direction, the cells on the cut surface show the extended cell shape rather than that in normal skin. In natural skin sample, the slicing angle was set parallel to the hair flow, which coincided with the tissue scale tension direction. As shown in Fig. 2a and b, fibroblasts in natural skin are less oriented than THS models, due to the tension direction in skin is complicated. It is possible that the cell nuclei of human skin were observed to be slightly smaller as a result of the deviation of the cell orientation direction and the section angle.

Specific comment 6. Figure 4f. For other HSE models it has been shown that there is a dose response to all-trans retinoic acid in terms of collagen synthesis – lower doses (250 – 500 nM) stimulated collagen production; higher doses inhibited it (Deshpande et al., Arch Dermatol Res. 2014). The concentration used here 10 micromolar is high. Was the response to different doses tested?

Our Response: In the report by Deshpande *et. al.*, *all-trans* retinoic acid was continuously administered to the dermal equivalent for 21 days. In our current study, we administered *all-trans* retinoic acid for 6-72 hours to HSEs consisting of epidermis and dermis. Our test conditions were analysed according to the reference to the report by Quan T *et. al.*, which was evaluated the function of *all-trans* retinoic acid on both human skin and HSE (Quan T *et al.*, Exp. Dermatol. 2011). We demonstrated that the results have mimic to that in human skin's reaction. Thus, we hope to remain this figure at the concentration of 10 μ M.

We hope that these changes meet with your approval. We greatly appreciate your comments, which provided a helpful perspective on our work.

Response to Reviewer #2

Comment: Very comprehensive and excellently written manuscript.

My major concern is that the authors over-emphasize the novelty of their findings. Most of these aspects have been demonstrated before, such as the suitability of skin models for skin irritation testing, that they can be used to assess the skin barrier function and to assess drug-related effects. Maybe it would be better to focus on the novel aspects (model to study tensional homeostasis). It would be helpful to include a bit more information about the differences between the models that have been used in this study.

Our Response: We are grateful for your careful suggestion for our manuscript. According to your comments, we revised the clearly and limited description of this model for applications (Line 35-37, 305-310, 377-380). The THS model and the TRS model are expected to contribute to the elucidation of the molecular mechanisms for skin physiological functions and pharmacological actions regulated by tensional homeostasis. We have also written the abstract and discussion to focus on novel aspects of the THS model for tensional homeostasis studies and the possibilities for novel screening system of quasi-drugs and pharmaceutical drugs under tensional homeostasis (Line 35-37, 101, 305-310, and 377-380).

Additional comments:

Additional comment 1. Fig. 4b: The Ki67 staining seems to be very unspecific. The IF images are not convincing.

Our Response: In accordance with this helpful comment, we have improved the images of Ki67 staining (Fig. 4d). The Ki67-positive cell ratio was calculated by counting the specific signal, which the ki67 staining signal is localized in the nucleolus, indicated by the arrow head in Fig. 4d.

Additional comment 2. Fig. 4g (also Fig. 5a): The changes in the gene expression levels are minor – this holds for several of the qPCR data.

Our Response: In the THS model, HAS3 gene expression in ATRA-contained medium treatment significantly induced a 3.7-fold compared to that in the control condition (Fig. 4g, left, IAM). In addition, topical application of HAS3 gene expression in the ATRA-contained lotion statistically significant induced a 4.9-fold compared to that in

the control condition (Fig. 4g, right, TAM). It has been shown that ATRA upregulates HAS3 gene expression in epidermal keratinocyte approximately 2.3-fold. Furthermore, in Fig. 5a, we have confirmed statistically significant differences and observed the increased intracellular localizations of ITGA2, MRTF-A, and α SMA by the immunohistochemical analyses (Fig. 5b). We think that these results would be physiologically reasonable. These data show that reproduced tensional homeostasis modulate mechanical stress signalling.

Additional comment 3. Fig. 4i: The control model (and also the other models) look pretty differentiated. Only very few, undifferentiated epidermal cells seem to be left. Overall, I don't see tremendous differences between the differently treated models. I feel it will be difficult to assess drug-related effects with that.

Our Response: As shown in the Material and Methods section, the test medium in which FGF, MAP, and FBS are depleted from the culture medium is used to evaluate functional molecules like the assays for cytokines *in vitro*. Because the additives have been removed, epidermal morphogenic function of the control group was suppressed. Such subtraction of culture additives is a general method in assessing cytokines, growth factors and healthcare materials.

Additional comment 4. Fig. 5c: I doubt that there is a statistically significant difference given the huge standard deviations and that the error bars are overlapping.

Our Response: We are grateful for your careful evaluation of our manuscript. In this analysis, the average value of MRTF-A signal intensity in the nucleus was calculated for $n=240$ nuclei and statistical analysis was performed (Line 553). Therefore, although the standard deviations look to huge and overlapping, a significant difference was obtained.

Additional comment 5. Page 14, line 217-19: I can't conclude this from the figures.

Our Response: In this sentence, we intended to emphasize that CK5-positive cells in the THS model more clearly form the single basal cell layer. In accordance with your comment, we performed analysis of expression pattern of type 17 collagen. The results showed more limited localisation to epithelial basal layer in the THS model compared to Bell' model. Following this result, we have modified the sentences in lines 215-219.

Additional comment 6. Page 14, line 224-226: The Ki67 staining appears to be highly unspecific. Can't draw this conclusion from the provided images

Our Response: In accordance with this helpful comment, we have improved the images of Ki67 staining (Fig. 4d). The Ki67-positive cell ratio was calculated by counting the specific signal, which the ki67 staining signal is localized in the nucleolus, indicated by the arrow head in Fig. 4d. As a result, we confirmed a statistically significant activation of cell proliferation in THS as compared to TRS and Bell's model.

We hope that these changes meet with your approval. We greatly appreciate your comments, which provided a helpful perspective on our work.

Response to Reviewer #3

We have studied your comments carefully and found that you understood the value and significance of our study in this field. We are grateful for your evaluation and valuable suggestions for our manuscript. Our specific responses are listed below:

Comment 1. Although the authors have touched on a few existing skin tissue equivalents in the introduction, their survey of literature is only limited on traditional transwell methods. Recently, a myriad of tissue engineering approaches have been used to recreate the skin tissues. Authors are encouraged to cite a few of recent papers that used microfluidics and 3D bioprinting to create bioengineered skin tissues.

Our Response: We are grateful to you for this suggestion. Application of tissue engineering approaches such as 3D bioprinting and microfluidic technologies to HSE is useful for high-level reproduction of complex 3D tissue structure and development of experimental multi-organ integrated model. We have added references for the tissue engineering approaches in the Introduction section (Line 88-90).

Comment 2. In line 116, the authors mention that Bell's model has been traditionally used for creating contracted dermal equivalents. However, they do not mention why there is a need for a new model, such as the one presented in this work. I encourage the authors to expand on the shortcomings of the Bell's method further and describe how their approach overcomes these shortcomings.

Our Response: We are grateful for your careful evaluation of our manuscript. Although many studies indicated that mechanical stimulation regulates skin physiological functions, the role of tensional homeostasis remains unclear. To our knowledge, this is the first study to generate a tissue equivalent model that can clearly evaluate the functionality of tissue-scale tensional homeostasis. The THS model and the TRS model will contribute to analyze for the molecular mechanisms of skin physiological functions and pharmacological actions regulated by tensional homeostasis as an alternative to animal experiments. According to your comments, we have revised that our description is suitable to understand the applicability for drug assessment compared to Bells model under tensional homeostasis in line 35-37, 101, 305-310, and 377-380.

Comment 3. What type and concentration of collagen was used to create the THS and

TRS models? Given that the skin ECM consists of many other proteins including elastin, the authors are encouraged to discuss the rationale behind using only collagen in their experiments.

Our Response: In accordance this helpful comment, we added the information in the Materials and Methods section (Line 412). We have used 4mg/ml bovine dermis-derived native collagen solution to construct our THS and TRS as well as the Bell's model. As pointed out by you, dermis consists of not only collagen but also elastin. Type 1 collagen is the major protein in skin and have been used as the most popular dermal matrix for construction of HSEs.

In order to reproduce more physiological tensional homeostasis, it is necessary to reconstruct the complex 3D dermal ECM structure composed of collagen fibers, elastic fibers and proteoglycans. Following your advice, we have added to the discussion the need for these investigations and will develop in further studies (Line 382).

Comment 4. The authors used mice skin tissues to generate the ex vivo models (Figure 1b). Why mouse tissue was used in this work? Skin contraction in mouse and human is different.

Our Response: In this experiment, the intact human skin should be fixed with a glass ring and excised in order to discuss tension balance in the physiological environment. We also think that human skin is ideal for more precise discussion, because the biophysical properties of mouse and human skin are different. However, as pointed by you, it was difficult to obtain suitable human skin for ethical reasons by basic scientists. We believe that there is a certain similarity between the mouse and human regarding the tensional homeostasis in the skin. We will study on your pointed subject with clinicians in the further studies.

We hope that these changes meet with your approval. We greatly appreciate your comments, which provided a helpful perspective on our work.

Reviewers' comments:

Reviewer #1 (Remarks to the Author):

I appreciate that the authors have replied to my concerns and queries, and addressed a few of them through new experiments and analyses. The replies and addition of data and information have provided some clarity. Yet, I feel that the main concern has in my opinion still not been addressed - the authors cannot claim from the presented data that the methodology/model they present significantly improves the characteristics of human skin equivalents. The superiority of this model in drug testing over e.g. Bell's model is not shown.

There are other concerns. Would not fixing a fibroblast-populated collagen gel to the edges of stiff wells, although the gel itself may from the beginning have properties approximating that of human dermis, in some areas overtime as the fibroblasts contract the gels result in supra-physiological tension? The contraction would be against a stiff edge rather than a material of similar stiffness of that of skin. Thus, would this not be more representative of the interphase between scar or fibrotic tissue and skin, rather than that of a skin continuum? The increased (epidermal) cell proliferation rate presented in figure 4d could partially be the result of increased tissue stiffening (e.g. Kenny FN et al., J Cell Sci 2018). The answer that the authors provide is not detailed enough to eliminate this concern.

The added collagen VII staining and collagen XVII staining of the THS are not convincing. The collagen VII staining is too diffuse and the collagen XVII staining at high magnification suggests improper polarization of basal keratinocytes. The diffuse collagen VII staining could be due to the antibody used, or be due to the short time of THS culture - it usually takes a few weeks to get collagen VII deposition, or reflect inherent differences of the THS model and skin. In terms of collagen VII staining the staining of Bell's model may be most convincing of the three models tested; although this staining is also fuzzy.

I noted that daily addition of ascorbic acid to the models is not mention in the Methods section. Has this mistakenly been omitted or was ascorbic acid not added? The latter could explain the generally imperfect collagen appearance.

Also the collagen I and III antibody used still appears to not specifically stain these collagens but rather produce pan-collagen staining.

Reviewer #2 (Remarks to the Author):

Major concern:

Fig. 4d: It seems that the authors just zoomed into the images and now present an excerpt/magnified picture. Even now, you can't conclude or see any differences in Ki67 expression between the three models. Still, it looks like there is distinct unspecific staining in Bell's model. Further, why is the Ki67 expressed by 100% for TRS? In human skin, Ki67 is exclusively expressed in basal cells and not even in every basal cell. Further, in skin equivalents, there shouldn't be any Ki67 positive cells above the basal cell layer and clearly not in all cells.

Further comments:

1. It would be helpful if the authors could add dotted lines to Fig. 4b, i, j to indicate the location of the epidermal and the dermal layer. Otherwise, that's difficult to get from the pictures, especially for non-skin experts.

2. Fig. 4b: The only difference I really see between the 3 skin models is in collagen expression.

The corresponding passage in the text should be amended accordingly.

3. I strongly suggest consulting a statistician. Student's t-test is not a suitable statistical method for these types of experiments (especially Fig. 4d) as a prerequisite for the Student's t-test is a normal distribution of the population/data.

Reviewer #3 (Remarks to the Author):

The authors have properly addressed all my comments. I do not have more comments for this study.

Response to Reviewer #1

We are grateful for your evaluation and valuable suggestions for our manuscript. Our specific responses are listed below:

Reviewer #1 (Remarks to the Author):

Comment 1: I appreciate that the authors have replied to my concerns and queries, and addressed a few of them through new experiments and analyses. The replies and addition of data and information have provided some clarity. Yet, I feel that the main concern has in my opinion still not been addressed - the authors cannot claim from the presented data that the methodology/model they present significantly improves the characteristics of human skin equivalents. The superiority of this model in drug testing over e.g. Bell's model is not shown.

Our Response: We are grateful for your careful evaluation of our manuscript. We have shown that tension balance affects responsiveness to several drugs such as ATRA and MAP. According to your comments, we have modified to a limited extent of the improvement of our model that our model is better to use for the optimization of tension load strength and verification of various molecular signals and drug responsiveness under the tension-loading condition rather than Bell's model. We have revised that our description is suitable understand the potential of THS model (Lines 35-36, 257-258, 303-307, and 374-376).

Comment 2: There are other concerns. Would not fixing a fibroblast-populated collagen gel to the edges of stiff wells, although the gel itself may from the beginning have properties approximating that of human dermis, in some areas overtime as the fibroblasts contract the gels result in supra-physiological tension? The contraction would be against a stiff edge rather than a material of similar stiffness of that of skin. Thus, would this not be more representative of the interphase between scar or fibrotic tissue and skin, rather than that of a skin continuum? The increased (epidermal) cell proliferation rate presented in figure 4d could partially be the result of increased tissue stiffening (e.g. Kenny FN et al., J Cell Sci 2018). The answer that the authors provide is not detailed enough to eliminate this concern.

Our Response: Thank you for your deep insight and comments on our manuscripts. As

pointed out by your criticisms, at this stage it is not possible to conclude whether the THS model reproduces normal or scarred skin, because we have not yet evaluated the tension of fibroblasts and ECM in natural skin and HSEs for ethical reasons. However, since fibroblast proliferation characteristic of wound healing is not observed in the fibroblasts of the THS model, we believe that it is at least different from scars and keloids. As you pointed out, it is possible to have intermediate properties from scars to normal skin. Although we also recognize that the optimization of traction force is an important issue, we are not able to perform the sufficient investigation at this time due to the limitation of the technology to optional adjust the tension of the tissue in cell culture condition. We will perform on your pointed subject with clinicians in the future studies.

Comment 2: The added collagen VII staining and collagen XVII staining of the THS are not convincing. The collagen VII staining is too diffuse and the collagen XVII staining at high magnification suggests improper polarization of basal keratinocytes. The diffuse collagen VII staining could be due to the antibody used, or be due to the short time of THS culture - it usually takes a few weeks to get got collagen VII deposition, or reflect inherent differences of the THS model and skin. In terms of collagen VII staining the staining of Bell's model may be most convincing of the three models tested; although this staining is also fuzzy.

Our Response: We thank you for your valuable comments on anchoring fibrils staining. The deposition of collagen VII and collagen XVII in the THS model seems to be incomplete compared to human skin (Fig.1 g). As you pointed out, the culture period may be too short for protein deposition. Due to the technical limitations of the maintaining tensional homeostasis and normal epidermal tissue phenotype, tissues at day 7 of air layer culture were analysed in this manuscript. In Fig. 4b, the expressions of collagen VII and collagen XVII were observed in all HSEs, and there seems to be no significant difference in the localization pattern. We have revised our manuscripts regarding collagen VII and collagen XVII staining (Lines 134-142, and 215-219). We will develop an additional novel THS model for long-term culture analysis in a future study.

Comment 3: I noted that daily addition of ascorbic acid to the models is not mention in the Methods section. Has this mistakenly been omitted or was ascorbic acid not added? The latter could explain the generally imperfect collagen appearance.

Our Response: The magnesium ascorbyl-phosphate (MAP) at the concentration of 1mM, which is an ascorbic acid derivative, is added to HSE every 48 to 72 hours, except for evaluation tests of functional molecules. We added the medium change frequency to the method section (Lines 423-424) in the revised manuscript.

Comment 4: Also the collagen I and III antibody used still appears to not specifically stain these collagens but rather produce pan-collagen staining.

As we received your helpful indication, we have performed the additional immunohistochemical analyses of collagen III by using another antibody. The updated human collagen III staining pattern was clearly different from collagen I and the specificity of antibody seems to be improved. Since the staining pattern of collagen I is similar to the previous reports (M EI-Domyati et al. Exp Dermatol. 2002), it seems that there is no problem with the antibody characteristics. Under this condition, collagen III was not detected in the THS model, which was constructed by using adult-derived fibroblasts, because this collagen is known highly produced and constructed during embryogenesis compared to adult stage (Valérie Haydont et al., Mech Ageing Dev. 2019). So we revised the expression regarding collagen expression in the manuscript (Line 134-142).

We hope that these changes will meet with your approval. Once again, we greatly appreciate that your comments were most helpful and gave us a better perspective of our work.

Response to Reviewer #2

We are grateful for your evaluation and valuable suggestions for our manuscript. Our specific responses are listed below:

Comment 1: Fig. 4d: It seems that the authors just zoomed into the images and now present an excerpt/magnified picture. Even now, you can't conclude or see any differences in Ki67 expression between the three models. Still, it looks like there is distinct unspecific staining in Bell's model. Further, why is the Ki67 expressed by 100% for TRS? In human skin, Ki67 is exclusively expressed in basal cells and not even in every basal cell. Further, in skin equivalents, there shouldn't be any Ki67 positive cells above the basal cell layer and clearly not in all cells.

Our Response: We are grateful for your careful evaluation of our manuscript. Following your suggestion, we have improved immunohistochemical staining condition and analyzed carefully. In the new staining image, non-specific staining in the stratum granulosum to the stratum corneum was improved and a specific signal of the nucleus was observed. In the previous analysis, the number of ki67-positive cells in the constant field of view of the THS model was set to 100%, and the values of the Bell's and TRS models were plotted. Following your suggestion, we calculated the ratio of ki67-positive cells to the total number of basal cells.

Specific comment 1: It would be helpful if the authors could add dotted lines to Fig. 4b, i, j to indicate the location of the epidermal and the dermal layer. Otherwise, that's difficult to get from the pictures, especially for non-skin experts.

Our Response: We are grateful for your helpful comment. According to your suggestion, we have added the dotted lines to Fig. 4b, d, i, j to indicate the location of the epidermal and dermal layers.

Specific comment 2: Fig. 4b: The only difference I really see between the 3 skin models is in collagen expression. The corresponding passage in the text should be amended accordingly.

Our Response: In Fig4b, the expression patterns of CK5, CK10, FLG, and CLDN1 in all THS models are similar to human skin, and there seems to be no significant

difference between the three models. However, COL7 and COL17-stainings seems to improve the polarity of the basal cell layer in the THS model compared to TRS and Bell's models, although the improvement is slight compared to human skin (Fig. 1g). We received your criticism, we have modified the sentences in lines 215-219.

Specific comment 3: I strongly suggest consulting a statistician. Student's t-test is not a suitable statistical method for these types of experiments (especially Fig. 4d) as a prerequisite for the Student's t-test is a normal distribution of the population/data.

As we received your helpful indication, we have revised all statistical analysis methods as follows. In Fig. 4d, we have changed the graph type from a bar chart to a box plot in order to show the distribution of samples, and reanalysis was performed by Dunnett's test which is a parametric multiple comparison procedure. Since HSE is constructed from uniform human primary culture cells, the distribution of ki67 positive cells is expected to be close to the normal distribution. Even in *in vivo* experiments, it seems that parametric analysis such as t-test was used to analyse the ki67 positive cell rate in previous reports (Eunjeong Kwon et al., Nat Commun. 2018, and Sarah Girardeau-Hubert et al., Sci Rep. 2019). In Fig. 4a, c, f, and g, and Fig. 5a, c, e, h, and i, reanalyses were performed on Dunnett's test or Tukey-Kramer test following two-way analysis of variance

We hope that these changes will meet with your approval. Once again, we greatly appreciate that your comments were most helpful and gave us a better perspective of our work.

REVIEWERS' COMMENTS:

Reviewer #1 (Remarks to the Author):

I would like to thank the authors for having carefully considered my comments. I apologize for not having noticed that MAP had been added to the medium.

In my opinion the clearer description of the model allowing the reader to better understand the potential of the model has improved the paper and eliminated my major concerns.

Reviewer #3 (Remarks to the Author):

The authors have replied to all reviewer's comments including my original comments. I have no other comments for the authors.